# PIVOT-CENTRIC TRAJECTORY PREDICTION: BRIDGING LONG HORIZONS VIA DYNAMICAL GUIDANCE

## ABSTRACT

Forecasting precise future motion of surrounding agents is essential for reliable autonomous vehicles. However, as the demand for longer prediction horizons increases, existing endpoint-completion or iterative-refine methods increasingly struggle with weak guidance and compounding errors. To tackle the long-horizon prediction challenge, we propose Pivot-Centric Trajectory Prediction (PCTP). By introducing "pivots" and focusing on predicting pivot points along extended trajectories, we divide the long-term prediction task into short-term sub-tasks at various scales. Specifically, PCTP decouples the long-term trajectory predicting process into two processes: pivot prediction and pivot-based trajectory refinement. The pivot prediction process aims to utilize global map context and agent-to-agent interactions to identify these "pivot points", while the pivot-based trajectory refinement process focuses on local map details and refines the short-term trajectory based on predicted "pivot points". Compared with existing methods, PCTP provides more intermediate guidance while reducing compounding errors. Moreover, PCTP is a flexible approach that can be integrated into most state-of-the-art trajectory prediction models. Experimental results show that PCTP improves the prediction accuracy of leading models on both Argoverse I and Argoverse II datasets with minimal impact on model size. Specifically, PCTP combined with QCNet outperforms all published ensemble-free methods on the Argoverse II leaderboard at submission. The code is available at: https://anonymous.4open.science/r/PCNet.

## 1 INTRODUCTION

Predicting the long-term future trajectory of surrounding agents is of crucial importance for autonomous driving systems to capture others' intentions and make trustworthy decisions (Hagedorn et al., 2023). However, as the forecast horizon extends, the behavior uncertainty and ambiguity increase significantly, from an expanding search space, compounding errors, and the complex interaction among heterogeneous information, including dynamic agent history and static map features (Hu et al., 2023; Sun et al., 2022; Tolstaya et al., 2021; Salzmann et al., 2020; Gao et al., 2020; Liang et al., 2020; Ngiam et al., 2021; Zhou et al., 2022).

To ease the long-term prediction uncertainty, the research communities have mainly focused on endpoint-completion (Gu et al., 2021; Shi et al., 2022; Cui et al., 2023; Ye et al., 2023; Aydemir et al., 2023) and iterative-refinement methods (Chai et al., 2019; Zhou et al., 2023; Jiang et al., 2023; Liu et al., 2024; Zhou et al., 2024), where the former focuses on utilizing prior-knowledge-based endpoint to guide the trajectory prediction for the purpose of reducing the search space, while the latter prefer to refine trajectory proposals iteratively to utilize detailed context information. (See *Fig.* 1 for more details).

However, the guidance provided by a simple anchor decreases significantly as the horizon length increases for prior-knowledge-based methods. For example, in a scenario where an agent's goal is to bypass an obstacle in front over the next six seconds, using only one endpoint behind the obstacle can cause ambiguity between left and right bypass behaviors. In contrast, iterative-refinement-based methods can avoid this ambiguity because the proposed trajectory predicted in the first stage provides strong guidance—however, the quality of proposals matters (Zhou et al., 2023). A wrong proposal would lead to further incorrect context for the fine-grained final predictions.

Based on the insight into how humans interact with other traffic participants, we propose a new trajectory decoding scheme called Pivot-Centric Trajectory Prediction (PCTP), trying to combine long-term intention prediction and short-term fine-grained action learning. For this purpose, we propose the concept of *pivots*, which stands for the key points in a trajectory. PCTP consists of the process of Pivot prediction and pivot-guided trajectory prediction. In the first stage, the model predicts several pivots, which represent the long-term goal, considering agent-to-agent interaction and cause map information. In the second stage, the model infers the fine-grained waypoints based on the controlling pivot point to further capture the local kinematics. Moreover, the first stage can be divided into hierarchical pivot prediction processes, where each level focuses on different time scales.

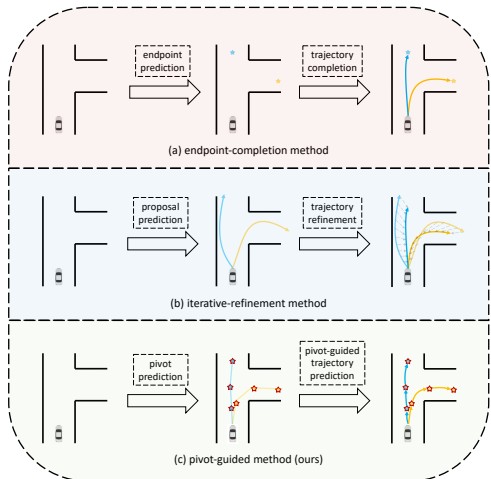

Figure 1: Illustration of various trajectory prediction methods: (a) The endpoint-completion method utilizes a single endpoint as the bridge, which may suffer from weak guidance. (b) The iterative-refinement method treats the entire trajectory as the bridge, leading to the accumulation of compounding errors. (c) Our pivot-guided trajectory prediction method provides sufficient global guidance while simultaneously minimizing compounding errors.

Compared to the single endpoint-completion method, PCTP offers more comprehensive intermediate guidance. For instance, these intermediate pivots provide clearer indications on when and where to make a right turn at an intersection, and on when and where to interact with other agents. Compared to the trajectory-refinement method, PCTP offers a much less intermediate process such that the cumulated noise and compounding errors could be minimized. The key insight is that high-quality pivots are easier to learn, even over longer horizons, due to the reduced search space compared to an entire trajectory. In PCTP, the factorization of pivot prediction and pivot-based trajectory refinement not only preserves the diversity and quality of anchors but also ensures that time-adjacent waypoints attend to the same local information. It is also worth noting that PCTP only changes the decoding strategy and does not introduce additional input, making it easy to integrate into most state-of-the-art trajectory prediction models. In summary, the key contributions of this work are as follows:

- We introduce Pivot-Centric Trajectory Evolution (PCTP), a method that infers trajectory based on predicted pivots, which factorizes global and local attention. Pivot prediction models intention through global interaction, while pivot-guided trajectory prediction focuses on local interaction.

- We propose a trajectory decoding framework based on PCTP, which can be integrated into any trajectory prediction model as a plugin.

- We evaluate the proposed method on the Argoverse I and Argoverse II datasets. Our method combined with QCNet outperforms all published ensemble-free works on the Argoverse II leaderboard.

## 2 RELATED WORK

Trajectory prediction involves taking scene representations, including surrounding agents' histories and road maps, as inputs to forecast the future motion of agents. There are two major learning-based approaches: dense representation and sparse representation. Dense representation methods Casas et al. (2018); Chai et al. (2019); Phan-Minh et al. (2020) use fixed-resolution grid structures to encode heterogeneous data and aggregate them using Convolutional Neural Networks, which focus on local information rather than global interactions. In contrast, sparse representation methods aggregate vector-based inputs using permutation-invariant set operators (Gao et al., 2020; Varadarajan et al., 2022), graph convolutions (Liang et al., 2020; Zeng et al., 2021; Da & Zhang, 2022; Liu et al., 2024; Liao et al., 2024), and attention mechanisms (Ngiam et al., 2021; Zhou et al., 2022; Nayakanti et al., 2023; Jiang et al., 2023; Philion et al., 2024), allowing for effective global in-

teractions. To address the challenge of long-term prediction uncertainty, recent work has adopted a two-stage scheme that factorizes the task into intention prediction and intention-based trajectory prediction.

### 2.1 ENDPOINT-BASED COMPLETION

Endpoint-based completion methods assume that a trajectory is largely defined by its endpoint. Endpoint can be classified from a pre-defined set Shi et al. (2022); Cui et al. (2023). TNT. Zhao et al. (2021) samples possible endpoints on centerlines in a dynamic scene and then predicts offsets from candidates to construct a discrete set for classification. DenseTNT. Gu et al. (2021) learns an endpoint distribution heatmap during training. MTR. Shi et al. (2022) clusters a set of endpoints in the dataset using K-Means before training, then uses these endpoints to query encoded features. Additionally, some methods Gilles et al. (2021; 2022) regress the endpoints using an anchor-free scheme.

### 2.2 TRAJECTORY-BASED REFINEMENT

Trajectory-based refinement methods Chai et al. (2019); Jiang et al. (2023); Liu et al. (2024) combine refinement networks Carion et al. (2020) from computer vision with trajectory prediction. They take a proposed trajectory generated in the first stage as input and predict the offset of each waypoint in the proposed trajectory. DCMS Ye et al. (2022) adds temporal and spatial constraints to the refinement network. QCNet. Zhou et al. (2023) embeds the proposed trajectory into a Fourier feature to query refinement-related features. SmartRefine. Zhou et al. (2024) uses an adaptive network to dynamically adjust refinement configurations and the number of iterations.

## 3 METHODOLOGY

PCTP is a two-stage trajectory decoding pipeline designed to alleviate the compounding error of long-term trajectory prediction, in which we follow a long-term pivot prediction and short-term trajectory refinement scheme. Fig.2 illustrates our framework. In *Sec.* 3.1, we provide a brief overview of the trajectory prediction problem. In *Sec.* 3.2, we delve into the intuition, design, and learning process underlying the pivot mechanism. *Sec.* 3.3 introduces pivot-guided trajectory prediction, while *Sec.* 3.4 presents the whole training details.

### 3.1 PROBLEM FORMULATION

Under the context of auto-driving, the goal of trajectory prediction is to predict future trajectories of surrounding agents given agent history and essential map context. Specifically, we have states of $N$ agents in the historic $T_h$ steps $\mathcal{A} = \{a_{-T_h+1}^{1 \sim N}, a_{-T_h+2}^{1 \sim N}, \cdots, a_0^{1 \sim N}\}$, and $M$ static local HD-map information $\mathcal{M} = \{m_1, m_2, \cdots m_M\}$. It is a common practice to encode raw data of various types into high-dimensional features using individual encoders:

$$\mathbf{e_a} = \mathcal{E_a}(\mathcal{A}) \in \mathbb{R}^{N \times T_h \times H}, \tag{1}$$

$$\mathbf{e_m} = \mathcal{E_m}(\mathcal{M}) \in \mathbb{R}^{M \times H}, \tag{2}$$

where $\mathcal{E_a}$ and $\mathcal{E_m}$ denote the agent encoder and map encoder, respectively. $H$ represents the features dimension after encoding.

Based on the encoder's output, we aim to predict $K$ possible future modalities for $N$ agents, where each modality is represented by the positions over $T_f$ future steps. The output $\mathcal{O}$ is typically generated by a decoder $\mathcal{D}$ as:

$$\mathcal{O} = \mathcal{D}(\mathbf{e_a}, \mathbf{e_m}) \in \mathbb{R}^{N \times K \times T_f \times p}, \tag{3}$$

where $p$ represents the dimension of each prediction, *e.g.*, $p = 2$ representing $(x, y)$. Usually, a $\mathcal{D}$ fuses information from $\mathbf{e}_a$ and $\mathbf{e}_m$, and output each agent' possible trajectories directly.

PCTP, as a decoder plugin, follows the aforementioned overall framework. At the same time, we argue that directly decoding long trajectories from the raw encoded feature may lose many details and lead to significant compounding errors. We modify the decoding process of $\mathcal{D}$ by introducing

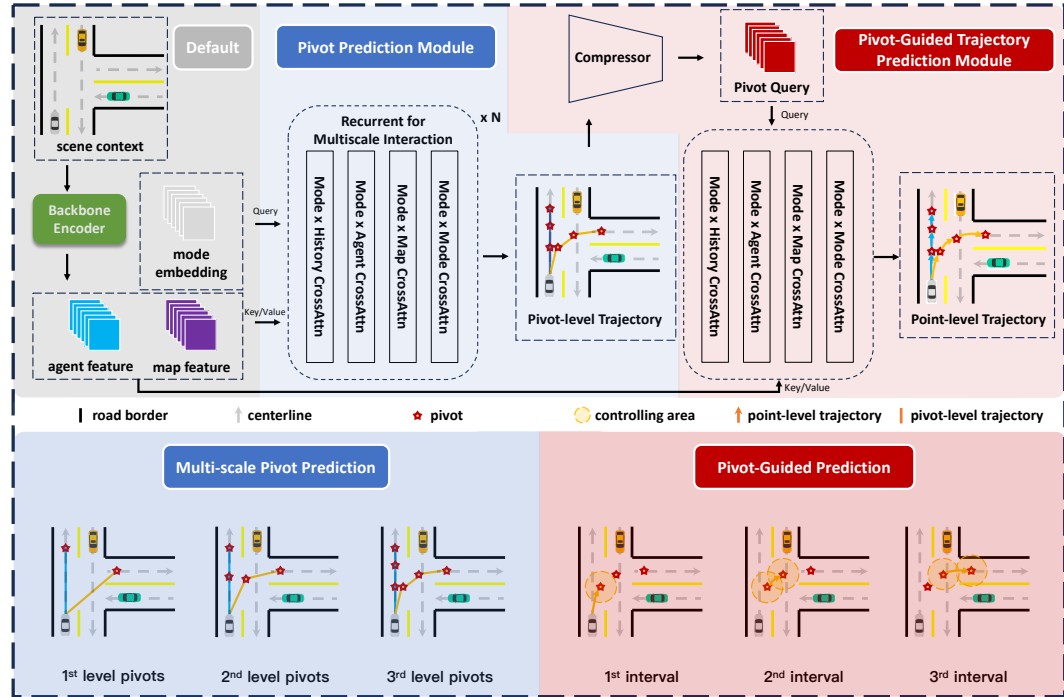

Figure 2: Overview of PCTP. The top section illustrates the overall pipeline, while the lower section details our Multi-Scale Pivot Prediction and Pivot-Guided Trajectory Prediction. First, the backbone encoder processes the HD map and agent-to-agent interaction information into a unified feature space. Next, the Pivot Prediction Module hierarchically predicts the positions of pivot points while simultaneously reducing the uncertainty associated with each pivot. Finally, the Pivot-Guided Trajectory Prediction Module decodes trajectories with the guidance of local pivots.

pivots, which represent these important intermediate key points. Then, the decoding process would be decoupled as Pivot Prediction(*Sec.* 3.2), and Pivot-Guided Trajectory Prediction (*Sec.* 3.3).

## 3.2 PIVOT PREDICTION

**Pivot** aims to bridge the gap between raw encoded features and the long-horizon predicted trajectories. Intuitively, we seek to identify the "pivot point" within trajectories that predominantly defines the trajectory temporal dynamics, thereby reducing distribution noise in the trajectory decoding process.

### 3.2.1 PIVOT DEFINITION

**Definition.** Intuitively, there are several ways to define pivots. Ideal pivots should consider map geometry, temporal dynamics, and agent interactions. However, it would be quite complicated and error-prone to design such complex pivots. In this paper, we propose that pivots focusing solely on temporal dynamics are sufficient for most scenarios. By definition, temporal-dynamic pivots are sampled from the raw trajectory, which could be defined by sampling from the ground truth as:

$$\{\mathbf{y}_1, \mathbf{y}_2, \cdots, \mathbf{y}_{T_f}\} \xrightarrow{\Delta_t \text{ sample}} \{\mathbf{y}_{\Delta_t}, \mathbf{y}_{2\Delta_t}, \cdots, \mathbf{y}_{T_f}\}, \tag{4}$$

where $\mathbf{y}_t$ represents the ground truth waypoint at $t$ time step. By introducing a time skip interval $\Delta_t$, pivots are defined by sampling from the raw ground truth. Fig 3 illustrates how we sample pivots from the initial trajectory. Using these pivots as the bridge between the raw feature and the final trajectory would significantly reduce unnecessary redundancy. Without loss of generality, we denote the sampled pivot points as $\mathcal{P} = \{p_0, p_1, \cdots, p_{T_f/\Delta_t}\}$.

**From End-Point To Pivot-Points.** In previous works (Gu et al., 2021; Shi et al., 2022; Cui et al., 2023; Ye et al., 2023; Aydemir et al., 2023), to reduce the trajectory uncertainty, a common practice involves predicting the endpoint of an entire trajectory in the first stage, with the intention of

dividing the whole trajectory action space into several endpoint-defined subspaces. This approach performs well for short-term trajectory predictions, where complex temporal dynamics are minimal. However, as the prediction horizon extends, the guidance provided by a single endpoint diminishes significantly, offering little improvement for long-term prediction tasks due to the lack of intermediate guidance. The endpoint acts like a distant destination in a fog: while the model knows where it needs to reach, there are no concrete steps to guide it along the way. We contend that, in the first stage, "pivot points" would aggregate more intermediate information and provide clear space-time transition paths.

**From Trajectory-Refine To Pivot-Refine.** In the long-term trajectory prediction task, there usually involves an extra stage as trajectory refinement that iteratively refines given trajectories to capture more fine-grained motion details. This extra refinement stage guidance significantly improves the model's ability to tackle long-term prediction tasks, which, at the same time, would suffer from more compounding errors when a longer time horizon is involved. In this paper, we argue that, in the second stage, a whole-size proposal trajectory is not necessary for the refinement, while pivots are all we need to conduct the refinement, as these pivots already contain all the necessary information for a target trajectory.

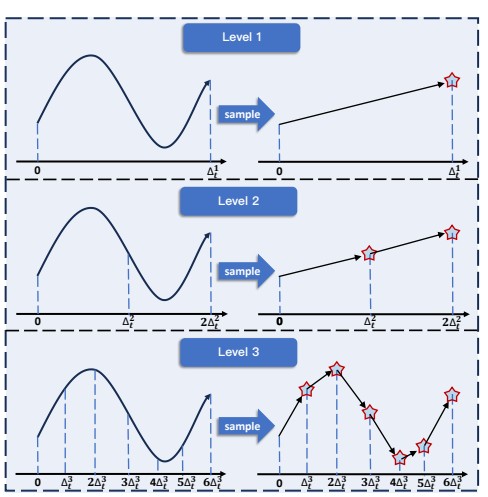

Figure 3: Sampling method for multi-scale pivots. The three point-level trajectories on the left represent the initial ground-truth trajectories, while the other three pivot-level trajectories represent the trajectories after pivot sampling. We use $\Delta_t^l$ as the sampling skip interval at level $l$ to sample pivots from the ground truth. The top section illustrates pivot sampling at the highest level, where only the endpoint is set as the pivot. As the number of levels increases, the skip interval decreases, allowing the model to focus on finer-grained features.

**Hierarchical Multi-Scale Pivots.** Humans typically begin with a high-level goal, then establish intermediate sub-goals, progressively decomposing these sub-goals into smaller, hierarchical objectives. Each level of goals focuses on different contextual aspects of the task. Drawing inspiration from this hierarchical cognitive process, PCTP further sets hierarchical multi-scale pivots as goals of different levels. We establish $L$ levels of pivots, where each level uses progressive sampling intervals. At the highest level, the interval is set to $T_f$, which means only the endpoint is considered as a *goal pivot*. Following, the second highest level interval is set to $T_f/2$, which means the endpoint and middle-point are considered as pivots, and this pattern continues downward. Formally, the hierarchical multi-scale pivots are defined by a set of hierarchical sampling intervals $\{\Delta_t^{(l)}\}$:

$$\Delta_t^{(l+1)} = \alpha_l \times \Delta_t^{(l)}, \qquad (5)$$

where $\Delta_t^{(l)}$ represents skip interval at the $l$-th level, $\alpha_l$ is interval growth factor from $l$-th level to $l + 1$-th level. Fig 3 illustrates how we sample pivots from different scales. Pivots sampled from different skip intervals certainly focus on different scale scene contexts. It is worth noting that pivots are predicted in agent-centric coordinates which can also be regarded as multi-scale relative spatial positional embedding.

### 3.2.2 PIVOT LEARNING

Pivot learning leverages information from $\mathbf{e_a}$ and $\mathbf{e_m}$ to decode pivots. Consistent with previous work Nayakanti et al. (2023); Varadarajan et al. (2022); Tang et al. (2024), we employ a cross-attention decoder to predict multiple groups of pivots with multiple learnable queries. Each group of pivots is decoded into a final single trajectory. Moreover, pivots are predicted using a hierarchical multi-scale schema that enables the decoder to progressively refine pivots based on high-level intentions and various scales of global context.

**Learnable Pivot.** Each query begins as a learnable embedding and corresponds to a final decoded trajectory in the interactive decoding process. In each level, queries cross-attend globally to the

scene context, and output pivots at the corresponding scale, representing the current level of intention. This mode-to-scene cross-attention involves three types of queries: mode-to-history, mode-to-map, and mode-to-agent queries. Through these specialized queries, each query retrieves essential features from the corresponding scene context, enhancing the overall trajectory prediction accuracy. Formally, given the encoded agent feature $\mathbf{e_a}$, map feature $\mathbf{e_m}$, and $K$ initial queries $\mathbf{q} \in \mathbb{R}^{K \times H}$, pivots are learned as:

$$\mathbf{e}_q = \mathrm{DAttn}\Big(\mathbf{q}, [\mathbf{e_a}, \mathbf{e_m}], [\mathbf{e_a}, \mathbf{e_m}]\Big), \tag{6}$$

$$\mathcal{P} = \mathcal{D}_{\mathrm{pivot}}(\mathbf{e}_q, \mathbf{e_a}, \mathbf{e_m}), \tag{7}$$

where DAttn, named Decoupled Attention, consists of mode-to-history attention, mode-to-agent attention, mode-to-map attention and mode-to-mode attention with $\mathbf{q}$ as the queries, $[\mathbf{e_a}, \mathbf{e_m}]$ as the keys and values. $\mathcal{P}$ represents the predicted pivots generated by the pivot decoder $\mathcal{D}_{\mathrm{pivot}}$.

**Multi-Scale Pivot Prediction.** Instead of introducing additional modules and parameters, we reuse the mode-to-scene cross-attention module to gradually refine the mode query from high-level intention to low-level pivots in an iterative manner. In each iteration, mode queries perform mode-to-scene cross-attention, outputting pivots at the current scale. Before entering the next iteration, the current output pivots are transformed to Fourier features, then embedded into the query feature space, and finally fused with the original query to achieve high-level intention embedding. Therefore, as iterations proceed, the pivot scale becomes finer, allowing for increasingly precise pivot refinement. Formally:

$$\mathbf{q}^{(0)} = \mathbf{q}, \quad \mathbf{q}^{(l)} = \mathcal{E}_{pivot}(\mathcal{P}^{(l-1)}), \ l > 0, \tag{8}$$

$$\mathbf{e}_q^{(l)} = \mathrm{MHA}\Big(\mathbf{q}^{(l)}, [\mathbf{e_a}, \mathbf{e_m}], [\mathbf{e_a}, \mathbf{e_m}]\Big), \tag{9}$$

$$\mathcal{P}^{(l)} = \mathcal{D}_{\mathrm{pivot}}(\mathbf{e}_q^{(l)}, \mathbf{e_a}, \mathbf{e_m}), \tag{10}$$

where we reuse the same pivot decoder $\mathcal{D}_{\mathrm{pivot}}$, and output multi-level pivots.

## 3.3 PIVOT-GUIDED TRAJECTORY PREDICTION.

In the second stage, we employ a Pivot-Guided Trajectory Prediction scheme to transform pivots into a fine-grained trajectory. Intuitively, each pivot offers substantial information within its neighborhood. Consequently, the Pivot-Guided Trajectory Prediction process operates by predicting the offset of each trajectory point from its associated pivot. This approach facilitates a hierarchical learning process, where the pivot captures macro dynamics while the trajectory learns the fine-grained temporal structure.

The process begins by using predicted pivots to query the essential local context around each pivot by pivot-to-context cross-attention. Given the predicted pivot $\mathcal{P} = \{p_0, p_1, \cdots, p_{T_f/\Delta_t}\}$, along with the encoded feature $\mathbf{e_a}$ and $\mathbf{e_m}$, the pivot embeddings are obtained by DAttn as follows:

$$\mathbf{e}_{p_i} = \mathrm{DAttn}\Big(\mathcal{E}_{pivot}(p_i), [\mathbf{e_a}, \mathbf{e_m}], [\mathbf{e_a}, \mathbf{e_m}]\Big). \tag{11}$$

Next, $\mathbf{e}_{\mathcal{P}_i}$ is used to predict the local offsets $\boldsymbol{\delta}_{p_i:p_{i+1}}$:

$$\boldsymbol{\delta}_{p_i:p_{i+1}} = \mathcal{D}_{\mathrm{traj}}(\mathbf{e}_{\mathcal{P}_i}, \mathbf{e_a}, \mathbf{e_m}), \tag{12}$$

$$\mathbf{o}_{p_i:p_{i+1}} = \boldsymbol{\delta}_{p_i:p_{i+1}} + p_i, \tag{13}$$

where $\mathcal{D}_{\mathrm{traj}}$ is our Pivot-Guided Trajectory Decoding Module, $p_i$ denotes the i-th pivot and $\mathbf{o}_{p_i:p_{i+1}}$ is final trajectory between $p_i$ and $p_{i+1}$. Under this framework, we reframe the initial long-term prediction task as a series of short-term subtasks, where each subtask operates within a more manageable temporal window.

In order to facilitate the integration of our model into most SOTA models, we provide two solutions corresponding to the two current mainstream trajectory decoding solutions.

**Pivot-Guided One-Shot Trajectory Decoding.** For one-shot trajectory decodersGu et al. (2021); Cui et al. (2023); Shi et al. (2022); Tang et al. (2024) which don't need iterative operations, PCTP only learns top-level pivots without iterative operations and then replaces mode-to-context cross-attention with pivot-to-context cross-attention, guiding the model to focus on relevant local context

information. After generating the trajectory, each waypoint is added to its assigned pivot, effectively performing short-term trajectory prediction at each pivot and then producing the final trajectory.

**Pivot-Guided k-Shot Trajectory Decoding.** For decoders with refinement Jiang et al. (2023); Zhou et al. (2023); Liu et al. (2024); Zhou et al. (2024), often referred to as k-shot trajectory decoding, PCTP also replaces mode-to-context cross-attention with pivot-to-context cross-attention, while retaining the iterative refinement module to achieve a more detailed trajectory. However, experimental results show that, with PCTP, the improvement gained from additional refinement becomes minimal. Consequently, the number of refinement iterations can be reduced to as few as one or even eliminated altogether.

## 3.4 TRAINING LOSS

Following common practice Zhou et al. (2022; 2023), we parameterize the pivots as a mixture of Laplace distributions:

$$f(\mathcal{P}) = \sum_{k=1}^{K} \pi_k \prod_{i=1}^{T_f/\Delta_t} \text{Laplace}(p_{i,k}|\mu_{i,k}, \mathbf{b}_{i,k}), \tag{14}$$

where $\{\pi_k\}_{k=1}^{K}$ are the mixing coefficients, and the Laplace density of the $k$-th mixture component for the $i$-th pivot is parameterized by location $\mu_{i,k}$ and scale $\mathbf{b}_{i,k}$. We optimize the mixing coefficients using a classification loss $\mathcal{L}_{\text{cls}}$.

In addition, we apply the Winner-Takes-AllLee et al. (2016) strategy to optimize PCTP. Specifically, our matching operation occurs during the pivot prediction stage to ensure intention consistency between pivot-level trajectory and point-level trajectory. This process is defined as follows:

$$k^* = \underset{k \in [1,K]}{\arg\min} \sum_{i=1}^{T_f/\Delta_t} \text{L2}(\mathbf{y}_{i \times \Delta_t}, \mathcal{P}_i), \tag{15}$$

where $\mathbf{y}_t$ denotes the ground truth at time step $t$ and $\mathcal{P}_i$ represents the $i$-th pivot. This formulation enables us to identify the most similar mode $k^*$ by minimizing the sum of L2 distances.

For stabilization, the pivot-guided trajectory decoding module stops the gradient flow through the pivots. The final loss function combines pivot-level trajectory loss $\mathcal{L}_{\text{pivot}}$, point-level trajectory loss $\mathcal{L}_{\text{traj}}$ and the classification loss $\mathcal{L}_{\text{cls}}$ for end-to-end training:

$$\mathcal{L}_{\text{pivot}} = \frac{\Delta_t^l}{T_f} \sum_{l=1}^{L} \mathcal{L}_{\text{preg}}^l, \tag{16}$$

$$\mathcal{L} = \mathcal{L}_{\text{pivot}} + \mathcal{L}_{\text{traj}} + \beta \cdot \mathcal{L}_{\text{cls}}, \tag{17}$$

where $\mathcal{L}_{\text{preg}}^l$ is the pivot's regression loss at $l$ level, $\Delta_t^l$ is the skip interval of $l$ level and $\beta$ is a hyperparameter that balances regression and classification. It is noted that the normalization factor $\Delta_t^l/T_f$ ensures that the learning rate for each pivot aligns with that for each point.

## 4 EXPERIMENTS

We show our result on validation set and test set of Argoverse I and Argoverse II. As shown in Table 1, PCTP can improve the accuracy of all considered state-of-the-art methods.

## 4.1 EXPERIMENTAL SETTING

**Datasets.** We evaluate our approach on two large-scale autonomous driving datasets: Argoverse I and Argoverse II. Both datasets capture real-world driving scenarios and include high-definition maps annotated with detailed motion data sampled at 10 Hz. Argoverse I contains 323,557 sequences collected from Miami and Pittsburgh. The prediction task requires forecasting 3-second future trajectories based on 2 seconds of historical observations. Argoverse II, comprising 250,000 scenarios across six cities, offers enhanced data quality and presents a more challenging task: predicting 6-second future trajectories given 5 seconds of observation history. We follow the official dataset guidelines for partitioning both datasets into training, validation, and test sets.

| Dataset | Method | b-minFDE$_6$ ↓ | minADE$_6$ ↓ | minFDE$_6$ ↓ | MR$_6$ ↓ | minADE$_1$ ↓ | minFDE$_1$ ↓ | MR$_1$ ↓ |
|---|---|---|---|---|---|---|---|---|
| Argoverse I | LaFormer | 1.759 | 0.918 | 1.091 | 0.096 | 1.473 | 2.810 | 0.474 |
| | LaFormer w/ Ours | **1.696** | **0.717** | **1.033** | 0.099 | **1.293** | **2.792** | 0.477 |
| | HPNet (no ref) | 1.580 | 0.663 | 0.933 | 0.078 | 1.370 | 2.940 | 0.465 |
| | HPNet (no ref) w/ Ours | **1.560** | **0.657** | **0.928** | 0.081 | **1.368** | **2.920** | **0.460** |
| Argoverse II | DenseTNT | 2.424 | 0.992 | 1.749 | 0.221 | 2.086 | 4.972 | 0.661 |
| | DenseTNT w/ Ours | **2.377** | **0.934** | **1.708** | **0.217** | **2.040** | **4.947** | **0.657** |
| | QCNet (no ref) | 1.928 | 0.729 | 1.292 | 0.164 | 1.680 | 4.348 | 0.590 |
| | QCNet (no ref) w/ Ours | **1.852** | **0.708** | **1.236** | **0.157** | **1.645** | **4.222** | **0.575** |
| | QCNet | 1.874 | 0.720 | 1.253 | 0.157 | 1.687 | 4.316 | 0.579 |
| | QCNet w/ Ours | **1.847** | **0.700** | **1.226** | **0.152** | **1.674** | **4.306** | 0.584 |

Table 1: Performance on Argoverse I and Argoverse II validation set. The sign (no ref) represents we implement the version that removes the refinement module based on the initial network since we try to prove that PCTP can be regarded as a lightweight refinement network, which leads to the effect of refinement being greatly reduced. PCTP improves most metrics of all state-of-the-art methods.

**Metrics.** Following the official evaluation protocols, we assess our approach using standard motion prediction metrics, including Brier minimum Final Displacement Error (b-minFDE$_K$), minimum Average Displacement Error (minADE$_K$), minimum Final Displacement Error (minFDE$_K$), and Miss Rate (MR$_K$). The minADE$_K$ metric calculates the $l_2$ distance between the ground truth trajectory and the best of $K$ predicted trajectories, averaged over all future steps, while minFDE$_K$ computes only the best prediction error among the $K$ predicted endpoints. For b-minFDE$_K$, we add $(1 - \hat{\pi})^2$ to minFDE$_6$ to evaluate classifier performance, where $\hat{\pi}$ denotes the highest predicted probability score output by the classifier. Additionally, MR$_K$ represents the proportion of predictions where minFDE$_K$ exceeds 2 meters. Typically, $K$ is set to 1 or 6; for cases where the number of trajectories exceeds $K$, we reduce the count by selecting those with the top-$K$ probability scores.

**Baselines.** The modular design of PCTP enables integration with most existing trajectory prediction frameworks. We demonstrate this versatility by evaluating PCTP with four state-of-the-art prediction models as backbones: HPNet Tang et al. (2024), LaFormer Liu et al. (2024), DenseTNT Gu et al. (2021), and QCNet Zhou et al. (2023).

## 4.2 QUANTITATIVE RESULT

**Performance on Val Set.** We present qualitative results on the Argoverse I and Argoverse II validation set in Table. 1. We integrate PCTP into each state-of-the-art prediction model and achieve good improvement. For instance, PCTP can reduce b-minFDE of LaFormer, HPNet, DenseTNT, and QCNet by 3.5%, 1.2%, 1.9%, and 3.9% respectively. For the implementation of HPNet and QCNet, we provide the result of both the original version and the version without the refinement module, since the pivot-guided module also has a refinement function.

**Performance on Test Set.** We also present some qualitative results on the Argoverse II test set in Table. 2. It shows that PCTP improves the results of QCNet by 0.3% on b-minFDE, 0.9% on minADE, 0.4% on minFDE, and 1.2% on MR. Specifically, PCTP based on QCNet outperforms all published ensemble-free works on the Argoverse 2 leaderboard.

| Method | b-minFDE$_6$ | minADE$_6$ | minFDE$_6$ | MR$_6$ |
|---|---|---|---|---|
| QCNet Zhou et al. (2023) | 1.861 | 0.636 | 1.241 | 0.154 |
| QCNet w/ Ours | **1.854** | **0.630** | **1.235** | **0.152** |

Table 2: Performance on Argoverse II test set. We choose QCNet as our baseline and submit the initial version and the version with PCTP.

## 4.3 ABLATION STUDY

To thoroughly evaluate the impact of each component of PCTP, we conducted ablation studies across three dimensions: module effectiveness, prediction performance across varying future time steps, and the effect of different pivot sampling intervals. We only show the performance of QCNet with PCTP, while the other results of baseline with PCTP can be found in the supplement materials.

**Component of PCTP.** We analyze the contribution of each core module in PCTP: Pivot Prediction(PP), Multi-Scale Pivot prediction(MSP), and Pivot-guided Trajectory Prediction(PTP). To isolate the effect of each module, we progressively remove them from the initial model and measure the resulting performance. Pivot learning provides foundational intermediate anchors, enhancing the model's ability to capture crucial motion dynamics. The multi-scale pivot refinement structure allows the agents' intention to adaptively interact at multiple scales, while the pivot-guided prediction module ensures the model can maintain temporal consistency by focusing on the local context around each pivot. Experimental in Table. 3 results demonstrate that each module contributes significantly to overall prediction accuracy, with the full configuration of PCTP achieving the highest accuracy.

| PP. | MSP. | PTP. | **b-minFDE$_6$** | minADE$_6$ | minFDE$_6$ | MR$_6$ |
|-----|------|------|------------------|------------|------------|--------|
| ✓ |  |  | 1.890 | 0.720 | 1.290 | 0.170 |
| ✓ | ✓ |  | 1.870 | 0.710 | 1.260 | 0.162 |
| ✓ | ✓ | ✓ | **1.847** | **0.700** | **1.226** | **0.152** |

Table 3: Ablation study results examining the contribution of three key modules: PP(Pivot Prediction), MSP (Multi-Scale Pivot prediction), and PTP(Pivot-guided Trajectory Prediction). The checkmark ✓ indicates the presence of a module. Experimental results are based on the Argoverse II validation set.

### 4.4 CASE ILLUSTRATION

We present a case illustration of PCTP integrated with QCNet in Fig 4. The PCTP framework generates 6 distinct trajectory modalities at the top level, each accompanied by a 6-second pivot prediction that explicitly specifies destination goals. As illustrated in Fig 4 (a), PCTP produces 4 straight-path modalities and 2 right-turn modalities. The straight modes include 1 deceleration mode (orange 'x') and 3 differentiated acceleration modes (red, gray, and purple 'x'), while the right-turn modes encode lane-specific intentions - first-lane (green 'x') and second-lane (blue 'x') targeting respectively. In subsequent stages, full trajectories are incrementally constructed through hierarchical refinement guided by these top-level pivot objectives in Fig 4 (b,c,d).

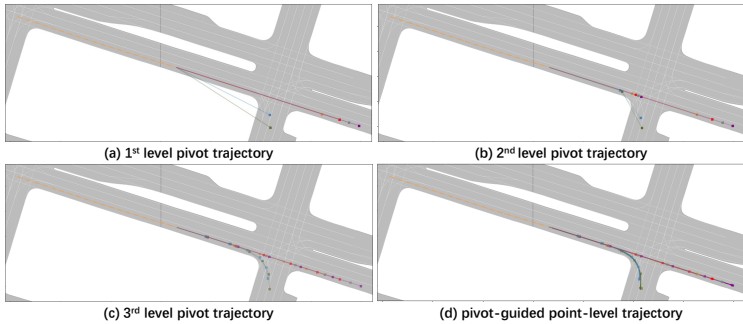

Figure 4: Visualization of PCTP learning process. Orange polyline denotes the agent history, while the other 6 different colors represent 6 modal trajectories respectively. As the pivot level increases from (a) to (d), the *right-turn* trajectory is gradually refined.

## 5 CONCLUSION

This paper introduces PCTP, a novel trajectory prediction framework, designed to address the limitations of traditional endpoint-based and iterative refinement methods, particularly for long-term prediction tasks. Unlike these two approaches, PCTP decomposes long-term trajectory prediction into a series of short-term sub-tasks. Considering hierarchical agent interactions and adaptive trajectory refinement, PCTP introduces pivot-level anchors that balance global intention with fine-grained local prediction. Ultimately, it significantly improves model accuracy and consistency over extended horizons. We believe that PCTP will serve as a foundation for future innovations in reliable trajectory prediction.

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

# A APPENDIX

## A.1 IMPLEMENTATION DETAIL

Since PCTP focuses on the trajectory decoding scheme, we experiment with encoders from several state-of-the-art models, including LaFormer Liu et al. (2024), HPNet Tang et al. (2024), DenseTNT Gu et al. (2021), and QCNet Zhou et al. (2023). We integrate PCTP into these backbones by modifying only the decoding scheme, without introducing additional attention mechanisms. For two-shot trajectory decoding methods, such as endpoint-based completion and iterative trajectory refinement, we adapt the models to predict the pivot-level trajectory in the first stage and the point-level trajectory in the second stage. Below, we provide detailed descriptions of our implementation.

### A.1.1 CROSS ATTENTION FOR MODE TO CONTEXT

We utilize cross-attention mechanisms that allow the mode query to interact with the scene context. To align with different models' decoding schemes, we reuse their cross-attention module respectively. Eg. for the QCNet version, we apply mode-to-agent, mode-to-history, and mode-to-map factorized attention, which QCNet takes in generating proposals; for the HPNet version, we use time query.

### A.1.2 PIVOT-LEVEL TRAJECTORY COMPRESSOR

We use a 2-layer MLP as a pivot-level trajectory compressor to aggregate the temporal and spatial information from pivots, whose hyperparameters are as follows:

- Number of layers: 2
- Input size: Number of pivots $\times$ 2
- Output size: 128

### A.1.3 DECODER

There are two decoders: pivots decoder and pivot-guided trajectory decoder. We use 2-layer MLP networks as well. The hyperparameters are as follows:

- Number of layers: 2
- Input size: 128
- Output size: Number of pivots or points $\times$ 2

### A.1.4 TRAINING

We train our models end-to-end using the AdamW optimizer, with the following configuration: 64 training epochs, 4 batch size per GPU, 8 A100 GPUs, 0.1 dropout rate, $1 \times 10^{-4}$ weight decay coefficient, 3 multi-scale levels, 1 pivot for 1st level, 2 pivots for 2nd level and 6 pivots for last level.

## A.2 EVALUATION METRIC DETAILS

**Minimum Final Displacement Error (minFDE)** measures the L2 distance between the endpoint of the best forecasted trajectory and the ground truth. The best here refers to the trajectory that has the minimum endpoint error.

**Minimum Average Displacement Error (minADE)** measures the average L2 distance between the best forecasted trajectory and the ground truth. The best here refers to the trajectory that has the minimum endpoint error.

**Miss Rate (MR)** count the number of scenarios where none of the forecasted trajectories are within 2.0 meters of ground truth according to endpoint error.

**Brier Minimum Final Displacement Error (b-minFDE)** is similar to minFDE. The only difference is that it adds $(1.0 - p)^2$ to the endpoint L2 distance, where $p$ corresponds to the probability of the best forecasted trajectory.

| Rank | Method | Published | b-minFDE$_6$ | MR$_6$ |
|------|--------|-----------|-------------|--------|
| 1 | LOF(ensemble) | ✓ | 1.63 | 0.12 |
| 2 | iDLab-SEPT++* | ✗ | 1.65 | 0.13 |
| 3 | EACON* | ✗ | 1.67 | 0.13 |
| 4 | PolarMotion* | ✗ | 1.71 | 0.12 |
| 5 | DeMo(ensemble) | ✓ | 1.73 | 0.12 |
| 6 | iDLab-SEPT* | ✗ | 1.74 | 0.14 |
| 7 | xPnC* | ✗ | 1.74 | 0.14 |
| 8 | XPredFormer* | ✗ | 1.76 | 0.15 |
| 9 | DyMap* | ✗ | 1.78 | 0.15 |
| 10 | RealMotion* | ✗ | 1.78 | 0.13 |
| **11** | **QCNet-AV2(ensemble)** | ✓ | **1.78** | **0.14** |
| 12 | MTC* | ✗ | 1.78 | 0.14 |
| 13 | tinymi* | ✗ | 1.80 | 0.16 |
| 14 | PolarMotion* | ✗ | 1.80 | 0.13 |
| 15 | AnonNet* | ✗ | 1.83 | 0.15 |
| 16 | JSS01* | ✗ | 1.84 | 0.15 |
| **17** | **QCNet w/ PCTP** | ✓ | **1.85** | **0.15** |
| - | SmartRefine | ✓ | 1.86 | 0.15 |
| - | QCNet | ✓ | 1.86 | 0.16 |
| - | ProphNet | ✓ | 1.88 | 0.18 |
| - | Gnet(ensemble) | ✓ | 1.90 | 0.18 |

Table 4: Argoverse 2 leaderboard at the time of paper submission. The works with ensemble tech are marked with (ensemble), and unpublished works are marked by * sign. The QCNet-AV2 is the ensemble version of QCNet, while the version without the ensemble is in the bottom section. In the bottom section, we show the top results of published work. Thus our method outperforms all published ensemble-free works on the Argoverse 2 leaderboard at the time of the paper submission.

| Model | history step | future step | skip interval | **b-minFDE$_6$** $\downarrow$ | minADE$_6$ $\downarrow$ | minFDE$_6$ $\downarrow$ | MR$_6$ $\downarrow$ |
|-------|-------------|-------------|---------------|-------------|-----------|-----------|--------|
| QCNet w/ Ours | 50 | 60 | 5 | 1.960 | 0.749 | 1.350 | 0.181 |
| QCNet w/ Ours | 50 | 60 | 10 | 1.970 | 0.752 | 1.370 | 0.186 |
| QCNet w/ Ours | 50 | 60 | 15 | 1.960 | 0.749 | 1.360 | 0.179 |
| QCNet w/ Ours | 50 | 60 | 20 | 1.930 | 0.745 | 1.350 | 0.183 |
| QCNet w/ Ours | 50 | 60 | 30 | 1.960 | 0.748 | 1.360 | 0.189 |
| QCNet w/ Ours | 50 | 60 | 60 | 2.000 | 0.765 | 1.410 | 0.201 |

Table 5: Ablation study on the time skip interval. We set the future step to 60 and change the skip interval to test the effect of the pivot number. Experimental results are based on the Argoverse II validation set.

### A.3 ABLATION STUDY ON TIME INTERVAL

As shown in Table. 5, for the experimental setting of 50 history frames and 60 future frames, we adjust different time intervals to get pivots of different scales, which can lead to the pivot-guided trajectory prediction module focusing on different local contexts. As we mentioned in the main paper, a small skip interval which equals to the endpoint-based-completion method and a large skip interval equaling to the iterative-trajectory-refinement method have defects respectively. PCTP can be regarded as a trade-off of both methods, which can guide models in different scales.

### A.4 ABLATION STUDY ON PREDICTION STEPS

As shown in Table. 6, we evaluated its performance across different prediction steps. This experiment tests whether PCTP offers greater improvements on longer future steps, where traditional methods tend to struggle. Since the total frames of a sample in Argoverse II are 110, we divide a sample into 20 history steps and 90 future steps. Then we set serval prediction tasks which contain 15, 30, 45, 60, 75, and 90 steps respectively. The experiment results in Table 6 show that the improvements achieved by PCTP increase as the prediction horizon lengthens, indicating that the

| Model | history step | future step | skip interval | $\textbf{b-minFDE}_6 \downarrow$ | $\text{minADE}_6 \downarrow$ | $\text{minFDE}_6 \downarrow$ | $\text{MR}_6 \downarrow$ |
|---|---|---|---|---|---|---|---|
| QCNet | 20 | 15 | - | 0.871 | 0.115 | 0.195 | 0.003 |
| QCNet | 20 | 30 | - | 1.150 | 0.293 | 0.493 | 0.024 |
| QCNet | 20 | 45 | - | 1.490 | 0.493 | 0.855 | 0.077 |
| QCNet | 20 | 60 | - | 1.980 | 0.766 | 1.340 | 0.183 |
| QCNet | 20 | 75 | - | 2.530 | 1.090 | 1.890 | 0.300 |
| QCNet | 20 | 90 | - | 3.110 | 1.470 | 2.480 | 0.427 |
| QCNet w/ Ours | 20 | 15 | 5 | 0.848 | 0.112 | 0.193 | 0.002 |
| QCNet w/ Ours | 20 | 30 | 5 | 1.100 | 0.284 | 0.480 | 0.021 |
| QCNet w/ Ours | 20 | 45 | 5 | 1.460 | 0.487 | 0.845 | 0.077 |
| QCNet w/ Ours | 20 | 60 | 5 | 1.920 | 0.755 | 1.310 | 0.178 |
| QCNet w/ Ours | 20 | 75 | 15 | 2.460 | 1.060 | 1.860 | 0.299 |
| QCNet w/ Ours | 20 | 90 | 15 | 3.090 | 1.450 | 2.470 | 0.427 |

Table 6: Ablation study on the length of the future horizon. We set the history step to 20 and change the future step to test the performance of PCTP. Experimental results are based on the Argoverse II validation set.

model effectively mitigates compounding errors over time. It is suggested that PCTP is particularly beneficial for long-term prediction tasks, where accurate trajectory refinement and guidance are essential.

## A.5 VISUALIZATION OF PCTP

Fig.5 and Fig.6 present visualizations of the predicted trajectories generated by our method. These results demonstrate that PCTP's pivot-level trajectory maintains greater modality diversity, while the pivot-guided point-level trajectory effectively focuses on local attention. In the second row of Fig. 5, we observe that the QCNet baseline fails to output the right-turn mode, which corresponds to the ground-truth trajectory. In contrast, QCNet integrated with PCTP successfully predicts this right-turn mode, as the pivot-level trajectory provides more diverse and higher-quality proposals. In the third row of Fig. 5, the QCNet baseline produces a trajectory that veers out of the drivable area. This occurs because the straight-mode trajectory heavily influences the proposal, and the refinement module struggles in such situations. On the other hand, while PCTP generates a similar proposal initially, the pivot-guided trajectory prediction leverages local attention to correct the final trajectory, ensuring it remains within the drivable area.

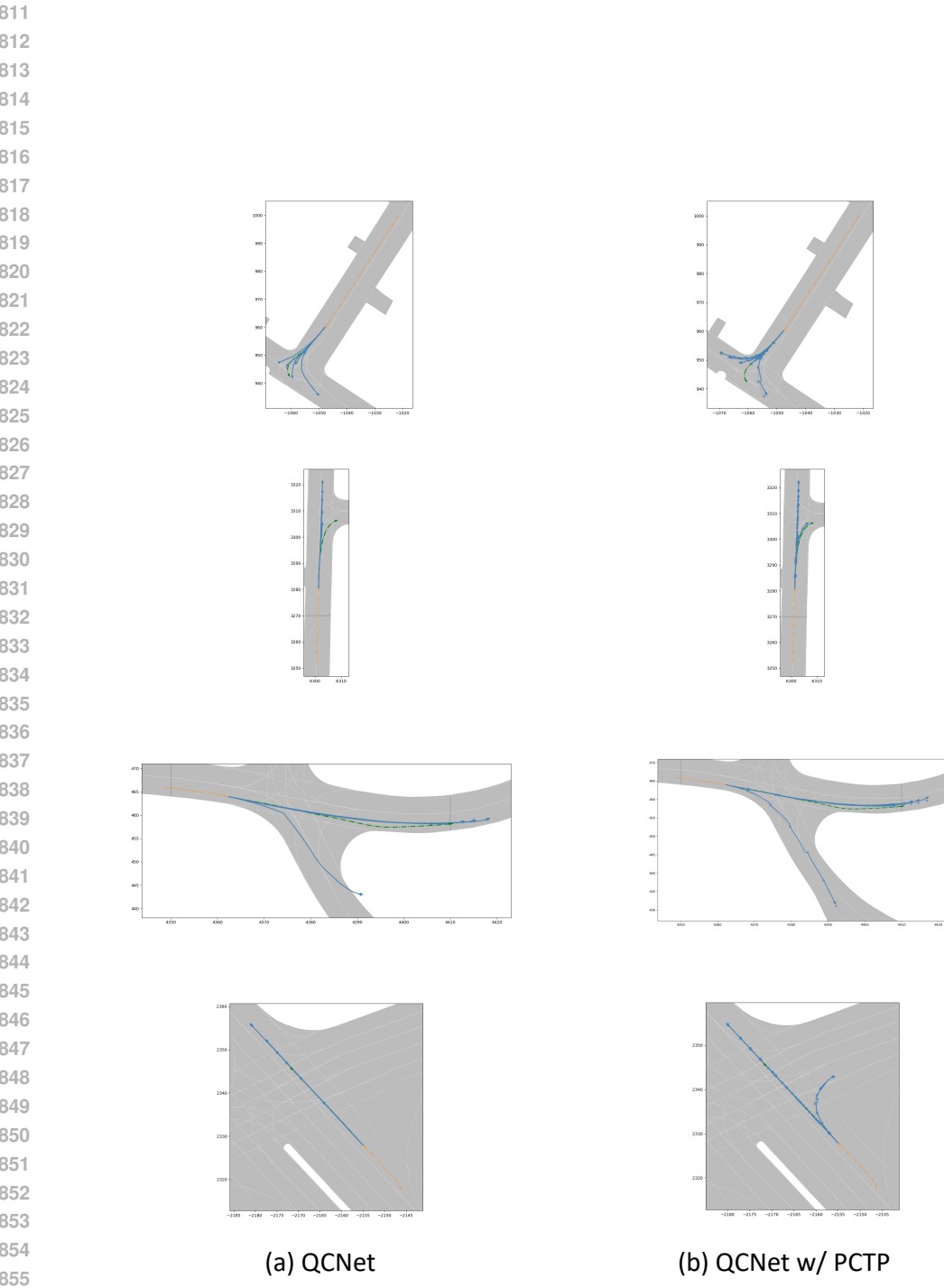

(a) QCNet                                        (b) QCNet w/ PCTP

Figure 5

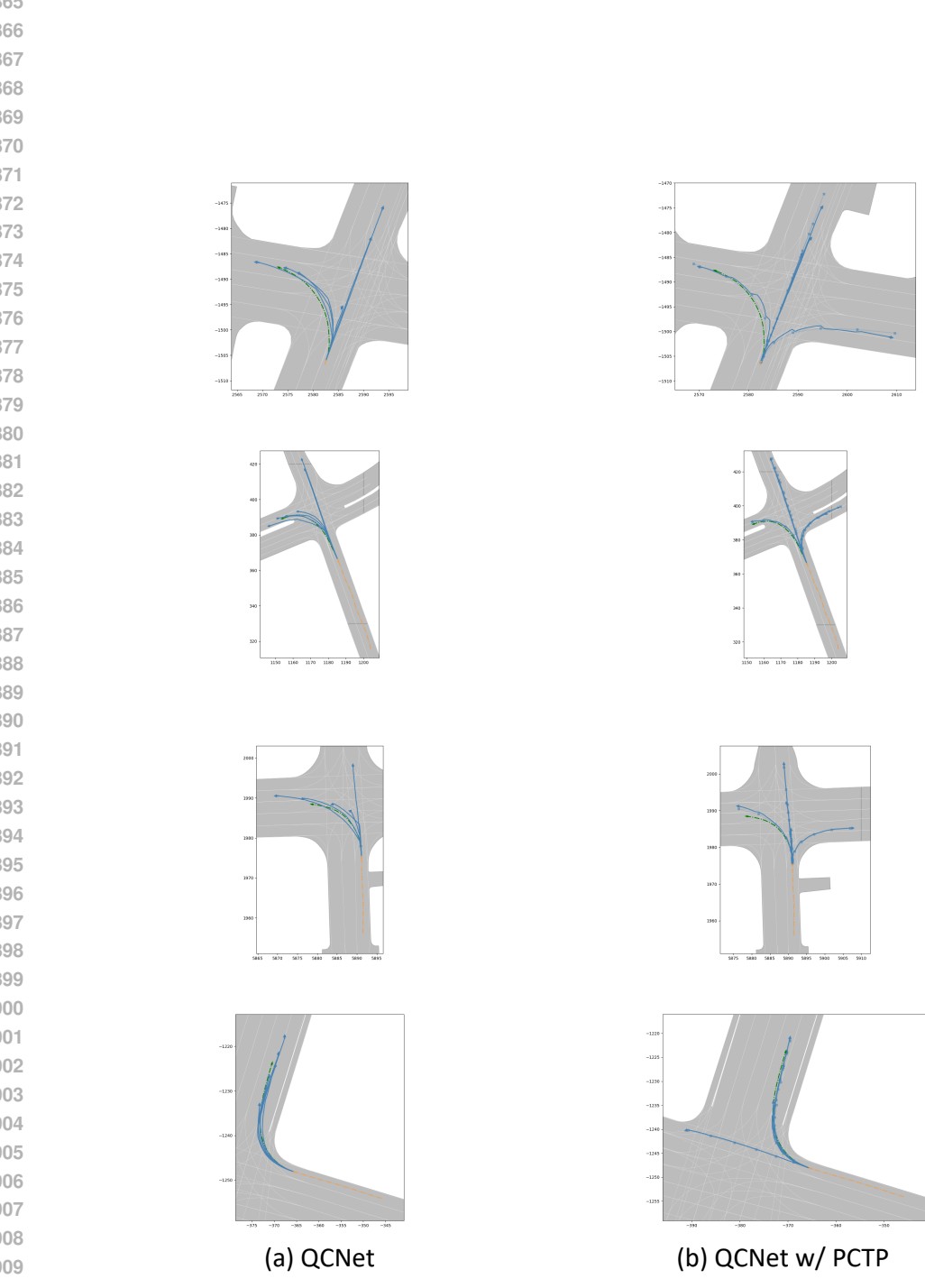

(a) QCNet                    (b) QCNet w/ PCTP

Figure 6