# OpenReview forum: "Pivot-Centric Trajectory Prediction: Bridging Long Horizons via Dynamical Guidance"
_ICLR.cc/2026/Conference — ICLR 2026 Conference Withdrawn Submission_

### Official Review · Reviewer_c3Kp · 2025-10-28

**Soundness:** 2
**Presentation:** 3
**Contribution:** 2
**Rating:** 2
**Confidence:** 4

**Summary:**

To address the long-horizon trajectory prediction problem, this work proposes Pivot-centric Trajectory Prediction (PCTP), which decomposes long-horizon trajectory prediction into two subtasks: pivot prediction and pivot-based trajectory refinement. The authors claim that compared with the endpoint-completion method, this approach can provide more guidance; compared with the iterative-refinement method, it can reduce compound errors. It achieves ensemble-free SOTA performance on Argoverse/Argoverse2 through flexible integration with existing models.

**Strengths:**

1.The paper is relatively clear in writing
2.As a decoding plugin, PCTP is decoupled from the main model architecture and has
demonstrated performance improvements across multiple mainstream models.

**Weaknesses:**

1.The proposed Pivot-guided method can be regarded as an intermediate form between the endpoint-guided method and the proposal-guided method. The authors argue that it can achieve the optimal balance between guidance and compound errors, yet it lacks theoretical derivation or more detailed experimental evidence to support this claim.

2.The proposed PCTP first refines pivots through multiple iterations and then completes the trajectory segment by segment. This is highly similar to SmartRefine, which first predicts proposals and then performs multiple iterative refinements—during each iteration, it adaptively selects anchor points to refine the corresponding trajectory segments.

Reference: SmartRefine: A Scenario-Adaptive Refinement Framework for Efficient Motion Prediction

3. The authors acknowledge that ideal pivots should account for map geometry, temporal dynamics, and agent interactions, but then state that focusing solely on temporal dynamics is sufficient for most scenarios. However, this claim seems somewhat assertive without sufficient evidence. It would strengthen the paper if the authors could provide either an ablation study or citation showing that incorporating map or interaction cues does not yield significant improvement. Otherwise, the statement risks sounding like a convenient simplification rather than a conclusion supported by analysis.

**Questions:**

1.Predicting pivots first and then completing the trajectory segment by segment may lead to poor trajectory continuity, which is also verified by the visualization results in the appendix.  Such discontinuous trajectories may have lower evaluation metrics, but it is obvious to humans that they are fake trajectories.  How do the authors view this issue?

2.From Table 6, the improvements brought by PCTP appear to be modest under different prediction horizons.  Have the authors attempted longer prediction horizons?  Will PCTP bring greater improvements under longer horizons compared to shorter ones?

3. Given that PCTP introduces hierarchical pivot prediction and trajectory refinement, what is the impact of PCTP on inference time? Are there specific inference speed data available to demonstrate its real-time performance in practical applications?

4. The paper's core insight is that high-quality pivots are easier to learn, even over longer
horizons, due to the reduced search space. How does the proposed method guarantee the highquality of these pivots? If the initial top-level pivot prediction is entirely incorrect, can the
subsequent refinement process still recover and produce an accurate trajectory?

---

### Official Review · Reviewer_abcE · 2025-10-30

**Soundness:** 3
**Presentation:** 3
**Contribution:** 1
**Rating:** 2
**Confidence:** 4

**Summary:**

This paper proposes Pivot-Centric Trajectory Prediction (PCTP), a two-stage decoding framework for trajectory prediction. The method predicts a small set of pivots and then performs local refinement between adjacent pivots. The paper claims that the proposed method provides stronger intermediate guidance than a single endpoint while avoiding compounding errors from long iterative refinement. By integration as a plugin into several backbones (LaFormer, HPNet, DenseTNT, and QCNet) , they show modest gains on Argoverse 1 and 2 Datasets. The best test-set result is a small improvement over QCNet on Argoverse 2.

**Strengths:**

1. On Argoverse 1 and 2 Datasets, adding PCTP to LaFormer, HPNet, DenseTNT, and QCNet improves b-minFDE, minADE and minFDE.
2. The paper does ablation study of each core module in PCTP: Pivot Prediction(PP), Multi-Scale Pivot prediction(MSP), and Pivot-guided Trajectory Prediction(PTP). By progressively removing them from the initial model, they show each module contributes significantly to overall prediction accuracy.
3. The presentation is clear and intuitive.
4. The appendix contains training setup and is useful for reproducibility.

**Weaknesses:**

1. The main issue with this paper is that the novelty is minimal. The idea of first predicting key frames and then predicting the entire trajectory based on key frames has previously been published. For example, see Paper [1] from Waymo in 2022. Paper [1] did experiments on both Argoverse Dataset and Waymo Open Motion Dataset (WOMD). Note that WOMD is more challenging than Argoverse in long horizon trajectory prediction, since WOMD requires predicting 8 seconds of future trajectories given 1 second of history, while Argoverse 1 requires predicting 3 seconds of future given 2 seconds of history, and Argoverse 2 requires predicting 6 seconds of future given 5 seconds of history. This paper claims that the method addresses long term prediction, but the evaluation omits WOMD, a widely used benchmark for long term prediction.
2. The improvement on the test set is quite marginal (Table 2).

In summary, because of the following reasons combined, this paper does not meet the standards of ICLR, and I would not recommend publication of this paper in ICLR conference:
1. The main idea of this paper has been published before.
2. There are no experiments on the WOMD dataset, an important benchmark for long term prediction.
3. The improvement on the Argoverse 2 test set is marginal.

[1] Lu, Qiujing, Weiqiao Han, Jeffrey Ling, Minfa Wang, Haoyu Chen, Balakrishnan Varadarajan, and Paul Covington. "Kemp: Keyframe-based hierarchical end-to-end deep model for long-term trajectory prediction." In 2022 International Conference on Robotics and Automation (ICRA), pp. 646-652. IEEE, 2022.

**Questions:**

Clearly comparing this paper with existing work on multi-waypoint or hierarchical prediction frameworks is a good start to improve this paper. In order to meet ICLR publication standards, the paper must articulate a clear novelty beyond these baselines and support it with solid experiments.

---

### Official Review · Reviewer_A9z9 · 2025-10-30

**Soundness:** 3
**Presentation:** 3
**Contribution:** 3
**Rating:** 6
**Confidence:** 3

**Summary:**

The paper introduces Pivot-Centric Trajectory Prediction (PCTP), a trajectory decoding framework designed to address the challenges of long-horizon motion forecasting. The method aims to find a middle ground between two dominant paradigms: endpoint-completion methods and iterative-refinement methods. PCTP's core idea is to introduce "pivots," a sparse set of key intermediate waypoints along a future trajectory. The framework operates in two stages: first, it hierarchically predicts these pivots at multiple time scales, starting from a coarse endpoint and progressively adding intermediate points. Second, it uses these final pivots to guide the generation of a fine-grained, complete trajectory by breaking the long-term task into a series of shorter-term sub-problems. The authors demonstrate that PCTP is a flexible, plug-in module that improves the performance of several state-of-the-art models on the Argoverse I and II datasets.

**Strengths:**

*   The paper is well-written and clearly articulates a significant and well-understood challenge in long-horizon trajectory prediction—the trade-off between guidance strength and the risk of error accumulation.
*   The concept of using sparse, learned "pivots" as an intermediate representation is an elegant and intuitive solution to this problem.
*   The design of PCTP as a modular, decoder-side plugin is a major strength, allowing it to be integrated with various existing encoder architectures. The experiments effectively demonstrate this versatility by showing consistent performance gains when PCTP is added to four different state-of-the-art models.

**Weaknesses:**

*   The definition of pivots is purely temporal, derived by sampling the ground-truth trajectory at fixed time intervals ($\Delta_t$). This approach may not be optimal, as it risks missing kinematically or geometrically critical points (e.g., the apex of a sharp turn, a necessary lane change) if they do not align with the sampling frequency. The paper acknowledges that ideal pivots would also consider map geometry and agent interactions but justifies the temporal-only approach as "sufficient" without a deeper analysis of its potential failure modes.
*   While the paper effectively contrasts PCTP with endpoint-completion and full-trajectory-refinement paradigms, its positioning relative to other hierarchical or coarse-to-fine methods that also use intermediate waypoints could be strengthened. The concept of using intermediate goals is not entirely new (see DESTINE: Dynamic Goal Queries with Temporal Transductive Alignment for Trajectory Prediction), and the paper's primary novelty lies in the iterative, multi-scale learning mechanism for these pivots. Emphasizing this distinction more clearly would better highlight the specific contribution.
*   The ablation study on the pivot sampling interval (Table 5) only analyzes the effect of the final interval at the most granular level. A more comprehensive study on the hyperparameters of the hierarchical structure itself—such as the number of levels (L) and the interval growth factor ($\alpha_l$)—would provide valuable insight into the model's robustness.

**Questions:**

1.  The paper argues that predicting sparse pivots is an easier learning task than predicting a full trajectory proposal. This is an intuitive claim. Is it possible to provide more direct quantitative evidence for this?
2.  The core novelty appears to be the multi-scale, iterative refinement of the pivots themselves. Could the authors elaborate on the specific advantages of this iterative process compared to other coarse-to-fine methods (e.g., DESTINE) that predict a set of intermediate waypoints in a single forward pass?
3.  In the multi-scale pivot prediction stage, the output pivots from a coarser level are used to form the query for the next, finer level. Could the authors provide more architectural detail on the embedding function ($\mathcal{E}_{pivot}$) and how the spatial information from the predicted pivots is transformed and fused with the existing query features to guide the subsequent refinement step?

---

### Official Review · Reviewer_2vTn · 2025-10-31

**Soundness:** 3
**Presentation:** 3
**Contribution:** 3
**Rating:** 6
**Confidence:** 4

**Summary:**

This paper proposes Pivot-Centric Trajectory Prediction (PCTP), a novel decoding strategy for long-horizon motion forecasting. PCTP introduces intermediate "pivot points" to bridge the gap between endpoint-completion and iterative-refinement methods. As a plug-in module, PCTP is shown to consistently improve the performance of several strong baseline models on the Argoverse 1 and 2 datasets, achieving state-of-the-art results.

**Strengths:**

1. Originality: The core concept of using hierarchical pivots as intermediate guidance is novel and well-motivated, offering a fresh perspective on structuring the trajectory decoding problem.

2. Quality: The work is technically sound and rigorously evaluated. Experiments on multiple baselines and datasets show clear, consistent improvements in key metrics. Ablation studies effectively validate the contribution of each component.

3. Clarity: The paper is exceptionally well-written. The problem is clearly introduced, the method is intuitively explained with helpful diagrams, and the visualizations effectively illustrate the benefits.

4. Significance: The proposed method is highly practical. Its plug-and-play nature and demonstrated performance gains make it a significant contribution with strong potential for adoption in both research and real-world applications.

**Weaknesses:**

1.	Novelty: It is intuitive to reduce the difficulty of model learning for trajectory prediction by introducing the "pivot points" for coarse-to-fine decoding. However, as I have not been following the latest progress, I am not sure if this approach was first proposed in this paper.
2.	Contribution: Since Trajectory Refine is a widely adopted trick in Trajectory Prediction, I hold that the most significant contribution of this paper lies in clearly demonstrating that "pivot points"-based Coarse-to-fine refinement offers greater advantages over Trajectory Refine. Could the authors directly illustrate this through the results? It is possible that such evidence has already been presented in the paper, so please forgive me if I may have overlooked this during my reading.
3.	Generalization: As long-horizon motion forecasting is the core application emphasized in the paper, could the authors further validate the generalization ability of the proposed scheme on the Waymo dataset, especially focusing on the results of 8-second prediction?
4.	Further improvement: Following MTR [1], have you tried to query map features from a "pivot points"-aligned local region?
5.	Further improvement: Following BiFF [2], have you tried to capture the interaction over "pivot points" from different agents for explicitly modeling future interaction?

[1] MTR: Motion Transformer with Global Intention Localization and Local Movement Refinement

[2] BiFF: Bi-level Future Fusion with Polyline-based Coordinate for Interactive Trajectory Prediction

**Questions:**

See Weaknesses

---

### Note · Authors · 2026-01-10

I have read and agree with the venue's withdrawal policy on behalf of myself and my co-authors.